# Salivary proteomics in monitoring the therapeutic response of canine oral melanoma

**Sekkarin Ploypetch[1,2], Sittiruk Roytrakul[3], Janthima Jaresitthikunchai[3], Narumon Phaonakrop[3], Patharakrit Teewasutrakul[2,4], Anudep Rungsipipat[2,5], Gunnaporn Suriyaphol** [1,2]*

**1** Biochemistry Unit, Department of Physiology, Faculty of Veterinary Science, Chulalongkorn University, Bangkok, Thailand, **2** Companion Animal Cancer Research Unit, Faculty of Veterinary Science, Chulalongkorn University, Bangkok, Thailand, **3** Functional Proteomics Technology Laboratory, Functional Ingredients and Food Innovation Research Group, National Center for Genetic Engineering and Biotechnology, National Science and Technology Development Agency, Pathum Thani, Thailand, **4** Oncology Clinic, Faculty of Veterinary Science, Small Animal Teaching Hospital, Chulalongkorn University, Bangkok, Thailand, **5** Department of Pathology, Faculty of Veterinary Science, Chulalongkorn University, Bangkok, Thailand

* Gunnaporn.V@chula.ac.th

**Data Availability Statement:** All relevant data are within the manuscript and its Supporting information files.

## Abstract

Saliva biomarkers are suitable for monitoring the therapeutic response of canine oral melanoma (COM), because saliva directly contacts the tumor, and saliva collection is non-invasive, convenient and cost effective. The present study aimed to investigate novel biomarkers from the salivary proteome of COM treated with surgery and a chemotherapy drug, carboplatin, 1–6 times, using a liquid chromatography–tandem mass spectrometry approach. The expression of a potential salivary biomarker, ubiquitin D (UBD), was observed and verified by western blot analysis. A significantly increased ratio of free UBD (fUBD) to conjugated UBD (cUBD) was shown in the pre-surgery stage (PreS) in OM dogs with short-term survival (STS) (less than 12 months after surgery) compared with that with long-term survival (more than 12 months after surgery). In dogs with STS, the ratio was also shown to be augmented in PreS compared with that after surgery, followed by treatment with carboplatin twice, 4 and 5 times [After treatment (AT)2, AT4 and AT5]. In addition, the expression of fUBD was enhanced in PreS compared with that of AT2 in the STS group. In conclusion, this study revealed that a ratio of fUBD to cUBD in PreS was plausibly shown to be a potential prognostic biomarker for survival in dogs with OM.

## Introduction

Canine oral melanoma (COM) is one of the most common head and neck tumors in dog [1]. The clinical staging system for the disease is classified into 4 stages as follows: stages I (a <2 cm diameter tumor) and II (a 2 to <4 cm diameter tumor), defined as early-clinical stages with no metastasis, whereas stages III (a ≥4 cm tumor and/or lymph node metastasis) and IV

**Funding:** This work was supported by the TRF Research Career Development Grant, RSA, (RSA5980053) to GS; the 100th Anniversary of Chulalongkorn University for a Doctoral Scholarship to SP; and the 90th Anniversary of Chulalongkorn University Scholarship to SP.

**Competing interests:** The authors have declared that no competing interests exist.

(a tumor with distant metastasis) are late-clinical-stage oral melanoma (LOM) [2]. Patients with LOM are most commonly found, owing to the difficulty in routinely examining tumors in dogs' mouths [1, 3]. They are usually treated with surgical resection in combination with the chemotherapy drugs carboplatin, doxorubicin or cyclophosphamide and piroxicam [4, 5]. Several factors can lead to failure of treatment, such as the nature of OM with high metastasis and high recurrence, owners' decisions not to pursue chemotherapy after surgical resection, and cancer drug resistance. Tumor biomarkers of the cancers might help owners make the appropriate decisions. Several tumor biomarkers have been used to help diagnosis, prognosis and surrogate endpoints, and monitoring treatment response and/or recurrence of the diseases [6]. It is noteworthy that mass spectrometry (MS)-based proteomics has been widely used to study novel expressed proteins in several cancers at a large scale, such as in tissues and saliva of COM, oral squamous cell carcinoma, benign tumors and chronic periodontitis, in tears of canine cancers, and in lymph nodes and serum of canine lymphoma [2, 7–10]. There remain knowledge gaps in proteome profiles of the COM therapeutic response. The objective of the present study was to investigate novel biomarkers from the salivary proteome of dogs with OM during pre-surgery (PreS), post-surgery (PostS) and after treating with the chemotherapy drug, carboplatin, for 1–6 times [After treatment (AT)1–AT6], using an in-solution digestion coupled with liquid chromatography–tandem mass spectrometry (LC–MS/MS). The candidate protein expression, ubiquitin D (UBD), was affirmed by western blot analysis.

## Materials and methods

### Animals

Saliva samples were collected from dogs with OM without previous treatment, either chemotherapy or radiotherapy. They were appointed for surgical excision and chemotherapy at the Small Animal Teaching Hospital, Faculty of Veterinary Science, Chulalongkorn University. The staging of OM was determined according to the World Health Organization [11]. Dogs were examined for an oral, regional lymph nodes and physical condition; moreover, the regional lymph nodes were required to rule out metastasis by cytological examination. Skull to abdomen radiography was evaluated by a Brivo DR-F Digital X-rays system (GE Healthcare, Little Chalfont, UK) or an Optima 64-slice helical CT unit CT-scan (GE Healthcare). Abdominal ultrasound was performed to detect OM metastasis to abdominal organs. Tumor diagnosis was achieved by cytology and histopathology.

Short-term survivors (STS) and long-term survivors (LTS) were defined as patients with late-clinical stage OM and living shorter than 12 months or longer than 12 months after surgery resection, respectively. Patient histories and patient treatment histories are shown in Tables 1 and 2, respectively. The study was approved by the Chulalongkorn University Animal Care and Use Committee (CU-ACUC), Thailand (Protocol No. 1631042). Written informed consents were obtained from all dog owners. All experiments were performed in accordance with the relevant guidelines and regulations.

### Sample preparation

Saliva was collected without mechanical or chemical stimulation as described previously [9]. Briefly, the patients were fasted and mouths were cleaned with 0.9% sterile normal saline solution before saliva collection. Samples were obtained at the initial visit for surgical excision (PreS group) and at 14 days after operation (PostS group). An adjuvant chemotherapeutic agent, carboplatin, was given at a dosage of 250 mg/m$^2$ at 3-week intervals for 6 or 7 treatments. Saliva was collected post chemotherapy treatments 1–6 times (AT1–AT6) and during follow-up at 1- or 2-month intervals 1–4 times after end of treatment (C1–C4). Saliva samples

**Table 1. Patient history.**

| Sample No. | ID | Age at initial treatment | Breed | Sex | Clinical stage | Survival time |
|---|---|---|---|---|---|---|
| 1 | 10 | 9 y 4 m | German shepherd | F | III | 8 m 23 d |
| 2 | 11 | 10 y 4 m | Poodle | F | III | 6 m 9 d |
| 3 | 16 | 11 y 5 m | Poodle | M | III | 3 m 19 d |
| 4 | 44 | 10 y 9 m | Mixed | Fs | III | 1 m 10 d |
| 5 | 46 | 7 y 7 m | Golden retriever | M | III | 3 m 22 d |
| 6 | 31 | 12 y 8 m | Shih tzu | F | III | 24 m 24 d |
| 7 | 71 | 12 y 3 m | Terrier | M | III | 14 m 12 d |
| 8 | 72 | 10 y | Mixed | Fs | III | 13 m 12 d |
| 9 | 86 | 13 y 1 m | Poodle | M | III | 15 m 21 d |

F, female; Fs, female spray; M, male.

were centrifuged at 2600 × g for 15 min at 4°C. Halt protease inhibitor cocktail (Thermo Fisher Scientific, Waltham, MA) was added and the supernatant was stored at −20°C until use.

## Preparation of saliva samples for LC–MS/MS analysis

Total protein of samples was measured by a modified Lowry protein assay [12]. Each sample was prepared to 1.5 µg/µL in 10 mM ammonium bicarbonate. Disulfide bonds were reduced by 10 mM dithiothreitol in 10 mM ammonium bicarbonate for 1 h at room temperature and alkylated in 100 mM iodoacetamide in 10 mM ammonium bicarbonate for 1 h at room temperature in the dark. After that, the protein in each sample was digested with the sequencing-grade modified trypsin (Promega, Madison, WI), using 50% acetonitrile (ACN) in 10 mM ammonium bicarbonate overnight. Then, the solvent was removed. Finally, each sample was dissolved with 20 µL of 0.1% formic acid and centrifuged at 10 000 rpm for 5 min before LC–MS/MS analysis. Spike bovine serum albumin (BSA) as internal standard was prepared by using 1.5 µg/µL in 10 mM ammonium bicarbonate.

## LC–MS/MS analysis and data processing

The samples were subjected to a reversed-phase high performance liquid chromatography (HPLC) system and an Ultimate 3000 LC System coupled to an HCTUltra PTM Discovery

**Table 2. Patient treatment history.**

| Sample No. | PreS | PostS | AT1 | AT2 | AT3 | AT4 | AT5 | AT6 | M1 | M2 | C1 | C2 | Remarks |
|---|---|---|---|---|---|---|---|---|---|---|---|---|---|
| 1 | • | • | • | • | • | • | • | | • | • | | | Metastasis |
| 2 | • | | | | • | • | • | • | | | • | • | Recurrence |
| 3 | • | • | • | | • | • | | | | | | | Recurrence |
| 4 | | • | • | • | | | | | | | | | Metastasis |
| 5 | | • | • | • | • | | | | | | | | Recurrence |
| 6 | • | • | | • | • | • | | • | | | • | • | Seizures |
| 7 | • | • | • | • | • | • | | | | | | | Metastasis |
| 8 | | • | • | • | • | • | • | | | | | | Metastasis |
| 9 | • | • | • | • | • | • | | | | | | | Recurrence |

PreS, pre-surgery; PostS, post-surgery; AT1–AT6, after treating with the chemotherapy drug for 1–6 times; M1 and M2, metastasis after treating with the last chemotherapy drug for 3 and 4 months, respectively; C1 and C2, check-up after treating with the chemotherapy drug 2 and 4 months without recurrence or metastasis.

System (Bruker Daltonics, Bremen, Germany) Peptides was applied to a PepSwift monolithic column (100 μm internal diameter × 50 mm) (Thermo Fisher Scientific) to separate with a linear gradient from 4% ACN, 0.1% formic acid (FA) to 70% ACN, 0.1% FA for 7.5 min with a regeneration step at 90% ACN, 0.1% FA and an equilibration step at 4% ACN, 0.1% FA at a flow rate of 1000 nL/min. It took 20 min per sample to complete the process. Peptide mass spectra were acquired in the positive ion mode with a scan range of 400 to 1500 m/z. However, peptides from the MS scan at 200–2800 m/z would be chosen if there were more than 5 precursor fragments. MS spectra data were analysed as described previously [13–15]. MASCOT software, version 2.2 (Matrix Science, London, UK), was used to search the peptide sequences against the NCBI mammal database for protein identification. Taxonomy (mammals), enzyme (trypsin), variable modifications (oxidation of methionine residues), mass values (monoisotopic), protein mass (unrestricted), peptide mass tolerance (1.2 Da), fragment mass tolerance (±0.6 Da), peptide charge state (1+, 2+ and 3+) and maximum number of missed cleavages were among the criteria used in the database search [14]. One or more peptides with an individual MASCOT score corresponding to p<0.05 were used to identify proteins, which were then annotated by UniProtKB/Swiss-Prot entries (http://www.uniprot.org/). jVenn diagram (http://bioinfo.genotoul.fr/jvenn/example.html) was used to exhibit the relationships among sample groups [16]. Stitch program, version 5.0 (http://stitch.embl.de/) was used to investigate the interaction network between potential proteins and anticancer drugs [17].

## Analysis of Western blots

To validate the MS results, 15 μg of samples were applied to a pre-cast NuPAGE 4–12% (w/v) Bis-Tris gel (Thermo Fisher Scientific) using NuPAGE MOPS SDS running buffer (Thermo Fisher Scientific) at 200 V for 60 min and PageRuler prestained protein ladder marker (molecular weight range 10–180 kDa) (Thermo Fisher Scientific). After that, the proteins were transferred to Trans-Blot Turbo nitrocellulose membranes (Bio-Rad Laboratories, Hercules, CA) at 25 V for 7 min using Trans-Blot Turbo 5× transfer buffer (Bio-Rad Laboratories) and total protein band intensities was detected using a Pierce Reversible Protein Stain Kit for Nitrocellulose Membranes (Thermo Fisher Scientific) as directed by the manufacturer. Blocking of nonspecific binding was achieved by 5% bovine serum albumin (GoldBio, St Louis, MO) in phosphate-buffered saline containing 0.1% Tween 20 (PBST) at 4˚C overnight. After being rinsed with PBST, a membrane was probed with 1:1000 dilution of primary antibodies at 4˚C overnight, including mouse monoclonal anti-human ubiquitin (Ub) (A-5) (Santa Cruz Biotechnology, Dallas, TX). Membranes were washed with PBST before being treated with 1:15 000 rabbit anti-mouse IgG secondary antibody coupled with horseradish peroxidase (Abcam, Cambridge, UK) for 1 h at 25˚C. ECL western blotting detection reagents (GE Healthcare) was utilized to visualized the target proteins. A ChemiDoc Touch Imaging System (Bio-Rad Laboratories) was used to image the chemiluminescent blots and Image Lab 6.0.1 software (Bio-Rad Laboratories) was used to analyse protein band intensities [14]. Total protein normalization was performed with the modification of Aldridge et al. (2008) [9, 18]. The intensities of the target bands (fUb and cUb) were compared to the total proteins in each lane. Western blotting was performed in triplicate.

## LC–MS/MS verification of expressed protein sequences

LC–MS/MS was used to validate Ub protein sequences from western blotting as stated previously [9]. Briefly, blotting membranes were incubated with Restore Plus Western Blot Stripping Buffer (Thermo Fisher Scientific) for 15 min prior to being rinsed 4 times with PBST. Excised protein bands were incubated with 10 mM dithiothreitol in 10 mM ammonium

bicarbonate overnight, trypsinized at 37˚C for 3 h and subjected to the LC–MS/MS as mentioned above [9, 14].

## Statistical analysis

ANOVA statistical analysis, incorporated into the DeCyder MS differential analysis software was used to analyse significantly different peptide peak intensities, whereas MASCOT software, version 2.2 was used to calculate MASCOT LC–MS/MS scores. Western blot analysis was proceeded by Kruskal–Wallis and Mann–Whitney tests for ratios of fUBD to cUBD, fUBD and cUBD expression. Kaplan–Meier survival curves were performed using log-rank (Mantel–Cox) and Gehan–Breslow–Wilcoxon analytic methods. GraphPad Prism, version 8.3.0 (GraphPad Software, La Jolla, CA) was used to statistically analyse protein expression data. Significance was accepted at the p<0.05 level [14].

## Results

Samples of LOM were divided into 2 groups according to the survival times: group 1, short-term survival (with median survival time less than 12 months) (STS); and group 2, long-term survival (with median survival time more than 12 months) (LTS). The STS and LTS groups had median survivals of 3 and 14.5 months, respectively. The two survival curves are illustrated in Fig 1 (p = 0.0046).

For the LC–MS/MS results, a total of 132, 29 and 74 proteins were commonly found in individuals of PreS, PostS, and metastasis (M), respectively (S1–S3 Tables). Two peptide fragments in all samples in every group of chemotherapy treatment, excluding the PreS, PostS, and M groups, appeared to be matched with the predicted UBD of *Rousettus aegyptiacus* and predicted transient receptor potential cation channel subfamily M (melastatin) member 8 (TRPM8) channel-associated factor 2 of *Monodelphis domestica*.

The expression of free UBD (fUBD) and conjugated UBD (cUBD) in STS and LTS samples was verifired by western blots and is illustrated in Figs 2 and 3 and S1–S3 Figs. The ratio of fUBD to cUBD and the fUBD levels were shown to be significantly augmented in the PreS group with STS compared with that with LTS (Tables 3 and 4). In dogs with STS, a significantly increased ratio of fUBD to cUBD was shown in PreS compared with that of AT2, AT4 and AT5 (Fig 2 and S1 and S2 Figs). In addition, the expression of fUBD was enhanced in PreS group with STS compared with that of AT2 (Table 2). The UBD sequence was confirmed by LC–MS/MS (Fig 4).

## Discussion

In the present study, LC–MS/MS and western blot analysis were used to identify novel salivary biomarker candidates of COM during pre-operation, post-operation, after treatments with carboplatin for 1–6 times, metastasis after treating with the last chemotherapy drug and check-up after treating with the chemotherapy drug without recurrence or metastasis. In-gel digestion coupled with mass spectrometric analysis (GeLC–MS/MS) and LC–MS/MS have been used to compare salivary proteomes of healthy dogs of different breeds, and healthy dogs with humans [19–22]. A previous publication reported the salivary proteome in dogs infected with *Leishmania infantum*, using LC–MS/MS [23]. For canine oral tumor proteomics, matrix-assisted laser desorption/ionization with time-of-flight mass spectrometry (MALDI-TOF MS) has been used to analyse peptide mass fingerprints, three-dimensional principal component analysis scatterplots and potential protein candidates in saliva and tissues of dogs with early-stage OM, late-stage OM, oral squamous cell carcinoma, benign oral tumors and healthy controls

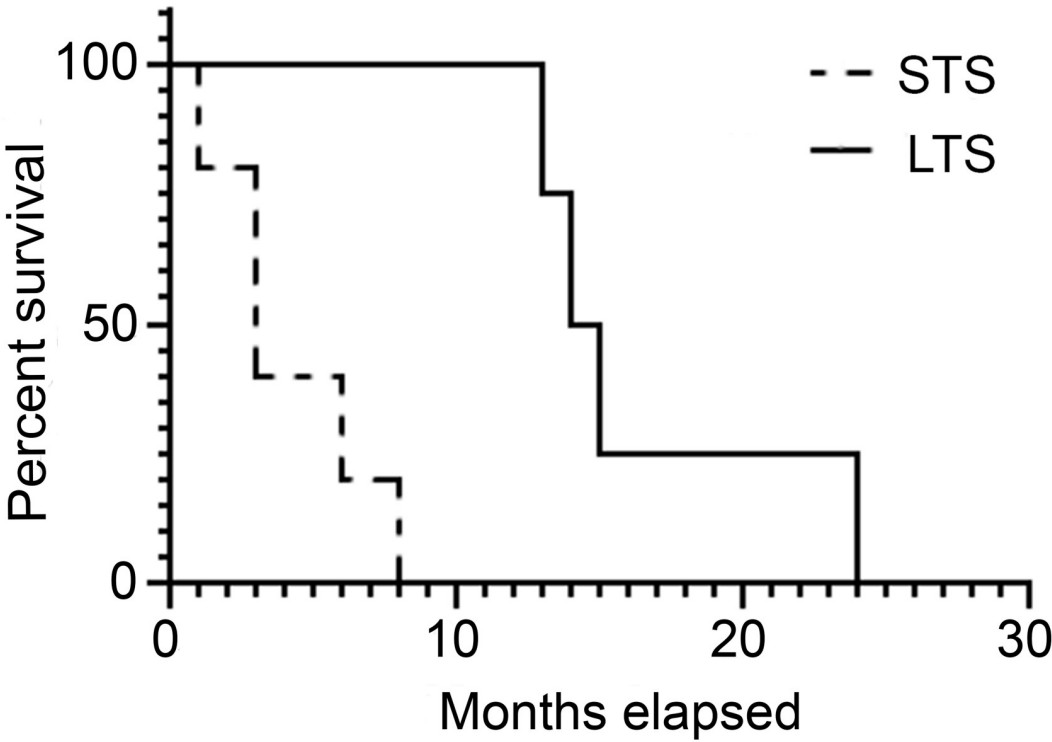

**Fig 1. Overall survival of patients with short-term survival (STS) (less than 12 months after surgery) and long-term survival (STS) (more than 12 months after surgery).**

[2, 9]. To the best of our knowledge, this study has shown for the first time the use of salivary proteomics of OM in dogs for monitoring surgery and chemotherapy responses.

From LC–MS/MS results, the expression of UBD appeared in all samples with chemotherapy treatment. The expression of UBD in different forms was further confirmed by western blot analysis. The expression of cUBD reflects the active ubiquitination. With the efficient ubiquitin conjugation reaction, reduced fUBD as well as the ratios of fUBD to cUBD would be observed. However, in the present study, ratios of fUBD to cUBD and the level of fUBD in PreS of LOM with STS were significantly higher than those in AT, in contrast with the trends of those of LOM with LTS, possibly showing the disable ubiquitination in STS and also exhibiting the potential prognostic biomarker for survival of LOM. Ubiquitin metabolism enzymes have been reported to link to cancer and several cancer-related signaling/regulatory pathways [24, 25]. Inhibition of the ubiquitin metabolic pathway associated with cancer growth, which was critical in the cancer treatment. In fact, ubiquitination (UBQ), the conjugation of ubiquitin to target proteins, leads to protein degradation by the 26S proteasome [26]. Bortezomib, a proteasome inhibitor that blocks the ubiquitin/proteasome pathway, has been approved for use in cancer treatment by the FDA [27]. Previous research examined free- and conjugated-ubiquitin expression in human colon carcinoma cell line to compare the efficacy of b-AP15, a new anticancer therapeutic target, with bortezomib. The expression of free- and conjugated-ubiquitins was shown to be associated with apoptotic proteins including p53, caspase-3, and cleaved PARP [28]. In patients with stage IIB–IIC colon cancer, the expression of UBD has been identified as a recurrent risk and associated with STS after surgery [29, 30]. UBD was also overexpressed in cervical squamous cell carcinoma tissues and associated with tumor size and lymphatic metastasis [31]. Silenced expression of UBD, regulated by miR-24-1-5p could

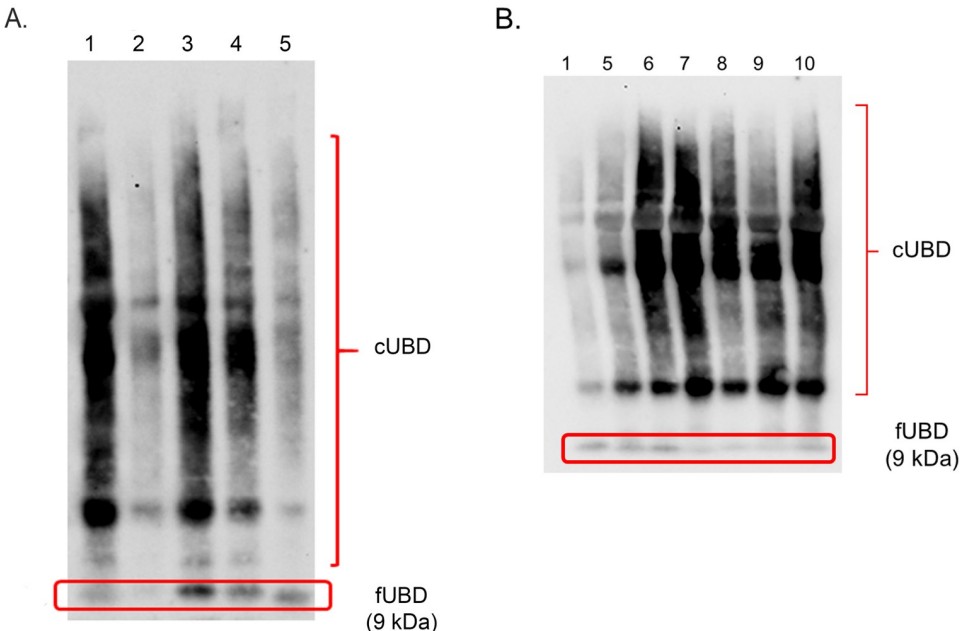

**Fig 2. Representative western blots of 2 patients (A and B) with short-term survival for free ubiquitin D (fUBD) at 9 kDa and conjugated ubiquitin D (cUBD) in saliva.** Lane 1, pre-surgery (PreS); lane 2, post-surgery (PostS); lane 3, after treating with chemotherapy drug once; lane 4, after treating with chemotherapy drug twice; lane 5, after treating with chemotherapy drug 3 times; lane 6, after treating with chemotherapy drug 4 times; lane 7, after treating with chemotherapy drug 5 times; lane 8, after treating with chemotherapy drug 6 times; lane 9, check-up after treating with chemotherapy drug 2 months; lane 10, check-up after treating with chemotherapy drug 4 months.

enhance autophagy and apoptosis of human skin melanoma cells [32]. In our study, according to the higher ratios of fUBD to cUBD in dogs with STS treated with chemotherapy, the increased cUBD after therapy in individuals with STS and the increased fUBD in individuals with LTS in the PreS group, it might be implied that the lower fUBD expression (or the higher UBQ) during treatment was associated with the STS. As several formulae of chemotherapy drugs and treatments have been used in treating canine oral cancers, other suitable drugs or treatments might be considered for treating the STS group with high ratios of fUBD to cUBD, with regard to the concept of precision medicine for canine oral cancer [4, 5, 33]. However, considering the limited sample sizes in the present study, the study should be continued in larger populations. In addition, misregulated expression of several ubiquitin-conjugating enzymes used in UBQ contributes to eccentric expression of nuclear factor κB (NF-κB) and transforming growth factor β and their signalling, leading to angiogenesis, increased invasiveness, chemotherapy resistance and metastasis of several cancers [34]. In fact, expression of NF-κB has been reported in the saliva of canine LOM and oral squamous cell carcinoma, and the expression of sentrin/small ubiquitin-like modifier-specific protease 7 (SENP7) has been reported in saliva of dogs with oral squamous cell carcinoma [9]. The link of NF-κB with ubiquitin should be investigated further.

## Conclusion

The present study has proposed for the first time a ratio of fUBD to cUBD in PreS as a candidate prognostic biomarker for survival in dogs with LOM. For future work, the expression of ubiquitin-conjugating enzymes used in UBQ and proteins involving in autophagy and

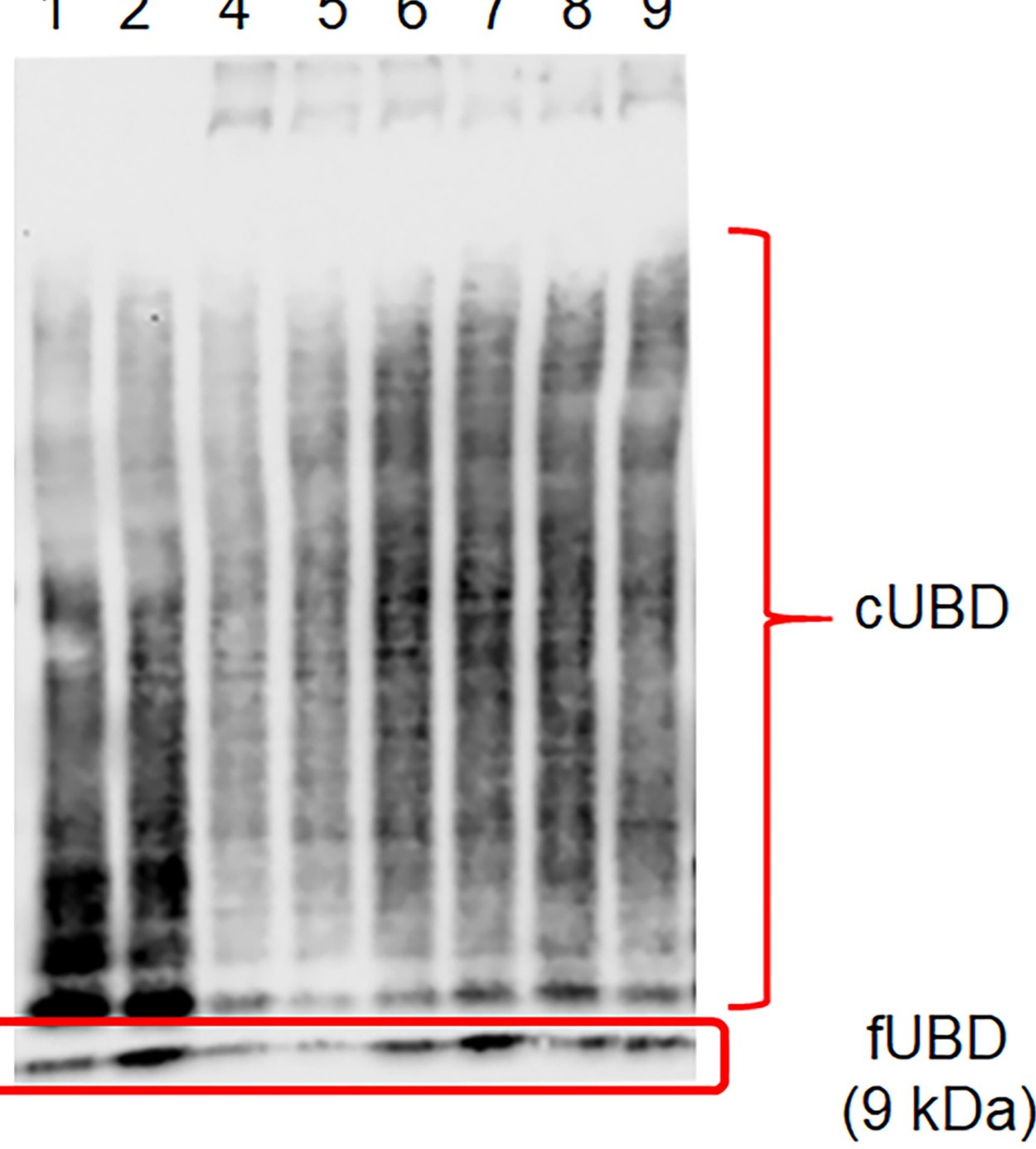

**Fig 3. Representative western blots of patients with long-term survival for free ubiquitin D (fUBD) at 9 kDa and conjugated ubiquitin D (cUBD) in saliva.** Lane 1, pre-surgery (PreS); lane 2, post-surgery (PostS); lane 3, after treating with chemotherapy drug twice; lane 4, after treating with chemotherapy drug 3 times; lane 5, after treating with chemotherapy drug 4 times; lane 6, after treating with chemotherapy drug 6 times; lane 7, check-up after treating with chemotherapy drug 2 months; lane 8, check-up after treating with chemotherapy drug 4 months.

**Table 3. Ratios of free ubiquitin D (fUBD) to conjugated ubiquitin D (cUBD) in patients with short-term survival (< 12 m after surgery) and long-term survival (> 12 m after surgery) at different time points.**

| Conditions | Ratio of fUBD to cUBD (Mean ± SD) | |
|---|---|---|
| | Short-term survival | Long-term survival |
| **1. PreS** | 4.89 ± 1.90[a, c, 1] | 1.92 ± 0.958[2] |
| **2. PostS** | 3.078 ± 2.09 | 3.30 ± 2.58 |
| **3. AT1** | 3.81 ± 2.86 | 7.27 ± 10.07 |
| **4. AT2** | 1.74 ± 1.24[b] | 4.40 ± 4.86 |
| **5. AT3** | 4.01 ± 3.89 | 6.22 ± 9.18 |
| **6. AT4** | 1.22 ± 0.50[d] | 11.16 ± 17.66 |
| **7. AT5** | 0.82 ± 0.55[d] | N/D |
| **8. AT6** | 2.84 ± 1.10 | 3.81 ± 1.94 |
| **9. M1** | 4.15 ± 2.58 | N/D |
| **10. M2** | 14.3 ± 1.14 | N/D |
| **11. C1** | 1.99 ± 0.463 | 8.70 ± 15.34 |
| **12. C2** | 1.32 ± 0.52 | 3.86 ± 1.33 |

[a,b] denote a significant difference at p<0.05.

[c,d] and [1,2] denote a significant difference at p<0.01.

PreS: Pre-surgery; PostS: Post-surgery; AT1−AT6: after treating with the chemotherapy drug for 1–6 times, respectively; M1 and M2: metastasis after treating with the last chemotherapy drug 3 and 4 months, respectively; C1 and C2: check-up after treating with the chemotherapy drug 2 and 4 months without recurrence and metastasis, respectively; N/D: not determined.

**Table 4. Free ubiquitin D (fUBD) and conjugated ubiquitin D (cUBD) in patients with short-term survival (< 12 m after surgery) and long-term survival (> 12 m after surgery) at different time points.**

| Conditions | fUBD (Mean ± SD) | | cUBD (Mean ± SD) | |
|---|---|---|---|---|
| | Short-term survival | Long-term survival | Short-term survival | Long-term survival |
| **1. PreS** | 0.14 ± 0.24[a, 1] | 0.26 ± 0.40[2] | 0.75 ± 0.13 | 0.80 ± 0.08 |
| **2. PostS** | 0.02 ± 0.02 | 0.03 ± 0.02 | 0.79 ± 0.14 | 0.81 ± 0.05 |
| **3. AT1** | 0.02 ± 0.01 | 0.05 ± 0.05 | 0.83 ± 0.10 | 0.69 ± 0.16 |
| **4. AT2** | 0.01 ± 0.01[b] | 0.16 ± 0.29 | 0.71 ± 0.27 | 0.82 ± 0.10 |
| **5. AT3** | 0.16 ± 0.31 | 0.27 ± 0.39 | 0.82 ± 0.09 | 0.85 ± 0.08 |
| **6. AT4** | 0.23 ± 0.45 | 0.24 ± 0.37 | 0.79 ± 0.19 | 0.64 ± 0.33 |
| **7. AT5** | 0.23 ± 0.45 | N/D | 0.90 ± 0.04 | N/D |
| **8. AT6** | 0.40 ± 0.55 | N/D | 0.71 ± 0.12 | N/D |
| **9. M1** | 0.03 ± 0.01 | | 0.69 ± 0.26 | |
| **10. M2** | 0.03 ± 0.03 | | 0.19 ± 0.19 | |
| **11. C1** | 0.38 ± 0.52 | | 0.70 ± 0.07 | |
| **12. C2** | 0.39 ± 0.54 | | 0.84 ± 0.09 | |

[a,b] and [1,2] denote a significant difference at p<0.05.

PreS: Pre-surgery; PostS: Post-surgery; AT1–AT6: after treating with the chemotherapy drug for 1–6 times, respectively; M1 and M2: metastasis after treating with the last chemotherapy drug 3 and 4 months, respectively; C1 and C2: check-up after treating with the chemotherapy drug 2 and 4 months without recurrence and metastasis, respectively; N/D: not determined.

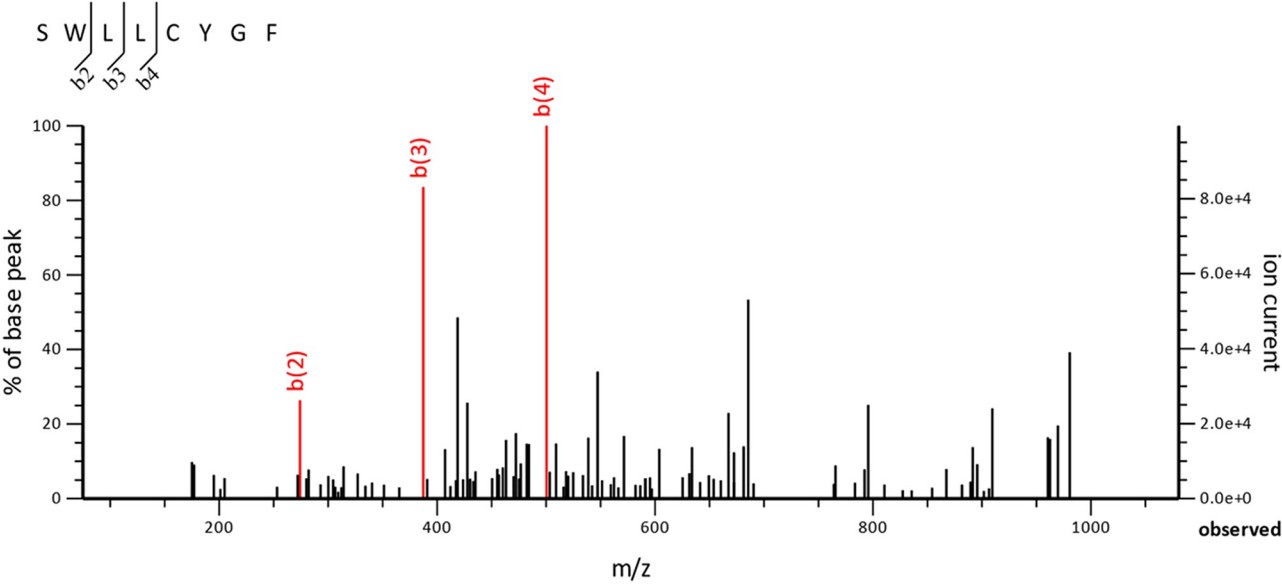

**Fig 4. Verification of ubiquitin sequence by liquid chromatography–tandem mass spectrometry (LC–MS/MS).** MS/MS fragmentation of **SWLLCYGF** found in free ubiquitin D (fUBD) is shown.

apoptosis should be investigated in larger populations. Suitable drugs or treatment might be reconsidered for treating the STS group with high ratios of fUBD to cUBD.

## Supporting information

**S1 Fig. Uncropped and unadjusted image of Fig 2A (patient No. 2).** Lane M, prestained protein ladder marker; lane 1, pre-surgery (PreS); lane 2, post-surgery (PostS); lane 3, after treating with chemotherapy drug once; lane 4, after treating with chemotherapy drug twice; lane 5, after treating with chemotherapy drug 3 times.
(TIF)

**S2 Fig. Uncropped and unadjusted image of Fig 2B (patient No. 3).** Lane 1, pre-surgery (PreS) of patient No. . . ..; lane 2, post-surgery (PostS); lane 3, after treating with chemotherapy drug once; lane 4, after treating with chemotherapy drug twice; lane 5, after treating with chemotherapy drug 3 times; lane 6, after treating with chemotherapy drug 4 times; lane 7, after treating with chemotherapy drug 5 times; lane P, positive control.
(TIF)

**S3 Fig. Uncropped and unadjusted image of Fig 3 (patient No. 6).** Lane M, prestained protein ladder marker; lane 1, pre-surgery (PreS); lane 2, post-surgery (PostS); lane 3, after treating with chemotherapy drug twice; lane 4, after treating with chemotherapy drug 3 times; lane 5, after treating with chemotherapy drug 4 times; lane 6, after treating with chemotherapy drug 6 times; lane 7, check-up after treating with chemotherapy drug 2 months; lane 8, check-up after treating with chemotherapy drug 4 months; lane 9, check-up after treating with chemotherapy drug 6 months; lane 10, check-up after treating with chemotherapy drug 8 months.
(TIF)

**S1 Table. Common proteins found in all individuals with OM during pre-surgery stage.**
(XLSX)

**S2 Table. Common proteins found in all individuals with OM during post-surgery stage.**
(XLSX)

**S3 Table. Common proteins found in all individuals with OM during metastasis.**
(XLSX)

**S1 Raw images.**
(PDF)

## Acknowledgments

We are grateful to Assoc. Prof. Chanin Kalpravidh, Dr. Worapan Tadadoltip and Ms. Suphansa Wanwattanakul for sample collection. Special thanks to Mr. Sucheewin Krobthong for helpful suggestions.

## Author Contributions

**Conceptualization:** Gunnaporn Suriyaphol.

**Data curation:** Sekkarin Ploypetch, Patharakrit Teewasutrakul, Anudep Rungsipipat.

**Formal analysis:** Gunnaporn Suriyaphol.

**Funding acquisition:** Sekkarin Ploypetch, Gunnaporn Suriyaphol.

**Investigation:** Sekkarin Ploypetch, Gunnaporn Suriyaphol.

**Methodology:** Sekkarin Ploypetch, Sittiruk Roytrakul, Janthima Jaresitthikunchai, Narumon Phaonakrop, Patharakrit Teewasutrakul.

**Project administration:** Gunnaporn Suriyaphol.

**Resources:** Sekkarin Ploypetch, Patharakrit Teewasutrakul, Anudep Rungsipipat.

**Software:** Sittiruk Roytrakul, Narumon Phaonakrop.

**Supervision:** Sittiruk Roytrakul, Gunnaporn Suriyaphol.

**Validation:** Sekkarin Ploypetch, Sittiruk Roytrakul, Gunnaporn Suriyaphol.

**Visualization:** Sekkarin Ploypetch, Anudep Rungsipipat, Gunnaporn Suriyaphol.

**Writing – original draft:** Sekkarin Ploypetch, Gunnaporn Suriyaphol.

**Writing – review & editing:** Sekkarin Ploypetch, Gunnaporn Suriyaphol.

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
