## [Decision Letter · Decision Letter 0]

7 Jul 2021

PONE-D-21-15429

Salivary proteomics in monitoring the therapeutic response of canine oral melanoma

PLOS ONE

Dear Dr. Suriyaphol,

Thank you for submitting your manuscript to PLOS ONE. After careful consideration, we feel that it has merit but does not fully meet PLOS ONE’s publication criteria as it currently stands. Therefore, we invite you to submit a revised version of the manuscript that addresses the points raised during the review process.

While the editor and reviewer were pleased with the overall work and found the work to be interesting and have potential for publication. We found that there is lack of details in the methods and analysis as well as in the discussion. Please see the detailed comments from the reviewer.

We look forward to receiving your revised manuscript.

Kind regards,

Sompop Bencharit, DDS, MS, PhD, FACP

Academic Editor

PLOS ONE

Additional Editor Comments:

While the editor and reviewer were pleased with the overall work and found the work to be interesting and have potential for publication. We found that there is lack of details in the methods and analysis as well as in the discussion. Please see the detailed comments from the reviewer.

Journal Requirements:

- https://bmcvetres.biomedcentral.com/articles/10.1186/s12917-020-02550-w

The text that needs to be addressed involves the Materials and Methods sections titled: LC-MS/MS data analysis, Western blot analysis, Verification of expressed protein sequences, and Statistical analyses.

In your revision ensure you cite all your sources (including your own works), and quote or rephrase any duplicated text outside the methods section. Further consideration is dependent on these concerns being addressed.

Reviewers' comments:

Reviewer's Responses to Questions

**Comments to the Author**

1. Is the manuscript technically sound, and do the data support the conclusions?

Reviewer #1: Yes

2. Has the statistical analysis been performed appropriately and rigorously? 

Reviewer #1: No

3. Have the authors made all data underlying the findings in their manuscript fully available?

Reviewer #1: Yes

4. Is the manuscript presented in an intelligible fashion and written in standard English?

Reviewer #1: Yes

5. Review Comments to the Author

Reviewer #1: This manuscript described the experimental study to utilize the ration of free ubiquitin D (fUBD)and conjugated ubiquitin D (cUBD) in saliva as biomarkers to relate the prognosis oral melanoma in dogs. The authors utilized saliva samples from 9 dogs at different stages of oral melanoma and subjected the samples to tyrpsing digestion and liquid chromatography-tandem mass spectrometry (LC-MS/MS) to verify the presence of the candidate protein including ubiquitin D. Western blot analyses were also used to determine the ration of fUGBD and cUBD.

Overall this review is a very interesting area of research to evaluate the potential features of ratio of fUBD and cUBD in saliva using Western blot analysis and identify the proteins using LC-MS/MS at the different stages of oral melanoma. The authors concluded that There are some questions that remain to be clarified as follows.

1. The figure 4 should be placed before figure 2 and 3 since the identification of UBD was performed before Western blot.

2. Could the author calculate the power analysis? What is the rationale of the sample size?

3. Is there any other way to quantify cUBD and fUBD protein levels besides Western blot analysis?

4. Could the author demonstrate the negative control and positive control in the Western blot analysis?

5. What is the expression profile of normal salivary UBD expression in a normal dog?

6. Would the level of UBD in saliva be affected by the saliva collection time?

7. Please discuss more details on the rationale to focus on the UBD expression in this study. Is there any literature on salivary UBD in other species related to cancer or melanoma? Please discuss.

Thank you very much.

6. PLOS authors have the option to publish the peer review history of their article (what does this mean?). If published, this will include your full peer review and any attached files.

Reviewer #1: **Yes: **Phimon Atsawasuwan

---

## [Author Response · Author response to Decision Letter 0]

20 Jul 2021

Response to the reviewers

We thank the editor and the reviewer for their thorough reviews and the constructive comments on the manuscript. As indicated below, we have checked all the comments provided by the editor and reviewers and have made necessary changes accordingly to their indications. 

Journal Requirements:

1. PLOS ONE now requires that authors provide the original uncropped and unadjusted images underlying all blot or gel results reported in a submission’s figures or Supporting Information files. 

- Done. The original uncropped and unadjusted images of Figures 2A, 2B and 3 are in S1-S3 Figures, respectively.

- Done. 

The text that needs to be addressed involves the Materials and Methods sections titled: LC-MS/MS data analysis, Western blot analysis, Verification of expressed protein sequences, and Statistical analyses. In your revision ensure you cite all your sources (including your own works), and quote or rephrase any duplicated text outside the methods section. Further consideration is dependent on these concerns being addressed.

- Done. Revised materials and methods are in a revised manuscript.

Reviewer's comments

1. The figure 4 should be placed before figure 2 and 3 since the identification of UBD was performed before Western blot.

- Figure 4 was from the second LC-MS/MS that we used to confirmed western blot results (Figures 2 and 3). Salivary proteins were applied to the first LC-MS/MS. After that we got a number of protein candidates and one of them was UBD. We performed western blotting of UBD and then excised a band from blotting membrane and confirmed the sequence of that band with the second LC-MS/MS. 

2. Could the author calculate the power analysis? What is the rationale of the sample size?

- According to the previous cross-sectional study of salivary proteomics of canine oral tumors, 24 LOM were used (Ploypetch et al., 2020). In the present study, we attempted to collect samples as many as possible as we realized that it would be pretty difficult to perform longitudinal study, using clinical samples (not laboratory animals). Likewise, we did not use the sample size calculator. In addition, various chemotherapy procedures are used in the animal hospital so some samples were excluded from this study. 

3. Is there any other way to quantify cUBD and fUBD protein levels besides Western blot analysis?

- Human UBD ELISA Kit is commercially available (MyBioSource, San Diego, CA). However, it could not separate cUBD from fUBD. In addition, some UBD antibodies used for western blots can be used for immunohistochemistry too.

4. Could the author demonstrate the negative control and positive control in the Western blot analysis?

- We have the positive control as shown in the figure below. The positive fUBD band at 9 kDa is shown in S2 Figure. However, we did not perform a negative control with no primary antibody which is a limitation of this study. 

5. What is the expression profile of normal salivary UBD expression in a normal dog?

- We could not find the normal levels of salivary UBD or ubiquitin in dogs. However, we found that various E3 ubiquitin-protein ligases were discovered in saliva of healthy dogs, using in-gel digestion combined with LC MS/MS and in-solution digestion coupled with LC MS/MS [1-4]. Salivary ubiquitin, polyubiquitin-B and polyubiquitin-C were upregulated in canine mammary tumors compared with healthy controls [2]. 

6. Would the level of UBD in saliva be affected by the saliva collection time?

- The dog patients in the present study were fasted prior to sample collection, surgery and/or chemotherapy. Hence, our samples would not be affected by time secretion or the environmental elements.

7. Please discuss more details on the rationale to focus on the UBD expression in this study. Is there any literature on salivary UBD in other species related to cancer or melanoma? Please discuss.

Lines 247-261: “..Ubiquitin metabolism enzymes have been reported to link to cancer and several cancer-related signaling/regulatory pathways [5, 6]. Inhibition of the ubiquitin metabolic pathway associated with cancer growth, which was critical in the cancer treatment. In fact, ubiquitination (UBQ), the conjugation of ubiquitin to target proteins, leads to protein degradation by the 26S proteasome [7]. Bortezomib, a proteasome inhibitor that blocks the ubiquitin/proteasome pathway, has been approved for use in cancer treatment by the FDA [8]. Previous research examined free- and conjugated-ubiquitin expression in human colon carcinoma cell line to compare the efficacy of b-AP15, a new anticancer therapeutic target, with bortezomib. The expression of free- and conjugated-ubiquitins was shown to be associated with apoptotic proteins including p53, caspase-3, and cleaved PARP [9]. In patients with stage IIB–IIC colon cancer, the expression of UBD has been identified as a recurrent risk and associated with STS after surgery [10, 11]. UBD was also overexpressed in cervical squamous cell carcinoma tissues and associated with tumor size and lymphatic metastasis [12]. Silenced expression of UBD, regulated by miR-24-1-5p could enhance autophagy and apoptosis of human skin melanoma cells [13]...” 

References

1. Bringel M, Jorge PK, Francisco PA, Lowe C, Sabino-Silva R, Colombini-Ishikiriama BL, et al. Salivary proteomic profile of dogs with and without dental calculus. BMC Vet Res. 2020;16(1):298. Epub 2020/08/21. doi: 10.1186/s12917-020-02514-0. PubMed PMID: 32814559; PubMed Central PMCID: PMCPMC7437026.

2. Franco-Martinez L, Tvarijonaviciute A, Horvatic A, Guillemin N, Bernal LJ, Baric Rafaj R, et al. Changes in saliva of dogs with canine leishmaniosis: A proteomic approach. Vet Parasitol. 2019;272:44-52. Epub 2019/08/10. doi: 10.1016/j.vetpar.2019.06.014. PubMed PMID: 31395204.

3. Ploypetch S, Roytrakul S, Phaonakrop N, Kittisenachai S, Leetanasaksakul K, Pisamai S, et al. In-gel digestion coupled with mass spectrometry (GeLC-MS/MS)-based salivary proteomic profiling of canine oral tumors. BMC Vet Res. 2020;16(1):335. Epub 2020/09/16. doi: 10.1186/s12917-020-02550-w. PubMed PMID: 32928212; PubMed Central PMCID: PMCPMC7489029.

4. Sanguansermsri P, Jenkinson HF, Thanasak J, Chairatvit K, Roytrakul S, Kittisenachai S, et al. Comparative proteomic study of dog and human saliva. PLoS One. 2018;13(12):e0208317. Epub 2018/12/05. doi: 10.1371/journal.pone.0208317. PubMed PMID: 30513116; PubMed Central PMCID: PMCPMC6279226.

5. Ciechanover A, Heller H, Elias S, Haas AL, Hershko A. ATP-dependent conjugation of reticulocyte proteins with the polypeptide required for protein degradation. Proc Natl Acad Sci U S A. 1980;77(3):1365-8. Epub 1980/03/01. doi: 10.1073/pnas.77.3.1365. PubMed PMID: 6769112; PubMed Central PMCID: PMCPMC348495.

6. Shi D, Grossman SR. Ubiquitin becomes ubiquitous in cancer: emerging roles of ubiquitin ligases and deubiquitinases in tumorigenesis and as therapeutic targets. Cancer Biol Ther. 2010;10(8):737-47. Epub 2010/10/12. doi: 10.4161/cbt.10.8.13417. PubMed PMID: 20930542; PubMed Central PMCID: PMCPMC3023568.

7. Fu H, Sadis S, Rubin DM, Glickman M, van Nocker S, Finley D, et al. Multiubiquitin chain binding and protein degradation are mediated by distinct domains within the 26 S proteasome subunit Mcb1. J Biol Chem. 1998;273(4):1970-81. Epub 1998/01/27. doi: 10.1074/jbc.273.4.1970. PubMed PMID: 9442033.

8. Nalepa G, Rolfe M, Harper JW. Drug discovery in the ubiquitin-proteasome system. Nat Rev Drug Discov. 2006;5(7):596-613. Epub 2006/07/04. doi: 10.1038/nrd2056. PubMed PMID: 16816840.

9. D'Arcy P, Brnjic S, Olofsson MH, Fryknas M, Lindsten K, De Cesare M, et al. Inhibition of proteasome deubiquitinating activity as a new cancer therapy. Nat Med. 2011;17(12):1636-40. Epub 2011/11/08. doi: 10.1038/nm.2536. PubMed PMID: 22057347.

10. Yan DW, Li DW, Yang YX, Xia J, Wang XL, Zhou CZ, et al. Ubiquitin D is correlated with colon cancer progression and predicts recurrence for stage II-III disease after curative surgery. Br J Cancer. 2010;103(7):961-9. Epub 2010/09/03. doi: 10.1038/sj.bjc.6605870. PubMed PMID: 20808312; PubMed Central PMCID: PMCPMC2965875.

11. Zhao S, Jiang T, Tang H, Cui F, Liu C, Guo F, et al. Ubiquitin D is an independent prognostic marker for survival in stage IIB-IIC colon cancer patients treated with 5-fluoruracil-based adjuvant chemotherapy. J Gastroenterol Hepatol. 2015;30(4):680-8. Epub 2014/09/23. doi: 10.1111/jgh.12784. PubMed PMID: 25238407.

12. Peng G, Dan W, Jun W, Junjun Y, Tong R, Baoli Z, et al. Transcriptome profiling of the cancer and adjacent nontumor tissues from cervical squamous cell carcinoma patients by RNA sequencing. Tumour Biol. 2015;36(5):3309-17. Epub 2015/01/15. doi: 10.1007/s13277-014-2963-0. PubMed PMID: 25586346.

13. Xiao Y, Diao Q, Liang Y, Peng Y, Zeng K. MicroRNA2415p promotes malignant melanoma cell autophagy and apoptosis via regulating ubiquitin D. Mol Med Rep. 2017;16(6):8448-54. Epub 2017/10/07. doi: 10.3892/mmr.2017.7614. PubMed PMID: 28983594.

Thank you very much.

---

## [Editor Report · Decision Letter 1]

2 Aug 2021

Salivary proteomics in monitoring the therapeutic response of canine oral melanoma

PONE-D-21-15429R1

Dear Dr. Suriyaphol,

We’re pleased to inform you that your manuscript has been judged scientifically suitable for publication and will be formally accepted for publication once it meets all outstanding technical requirements.

Kind regards,

Sompop Bencharit, DDS, MS, PhD, FACP

Academic Editor

PLOS ONE

Additional Editor Comments (optional):

Thank you for addressing all comments.
---

## [Editor Report · Acceptance letter]

9 Aug 2021

PONE-D-21-15429R1 

Salivary proteomics in monitoring the therapeutic response of canine oral melanoma 

Dear Dr. Suriyaphol:

I'm pleased to inform you that your manuscript has been deemed suitable for publication in PLOS ONE. Congratulations! Your manuscript is now with our production department. 

Kind regards, 

on behalf of

Dr. Sompop Bencharit 

Academic Editor

PLOS ONE